# Triggered chain reaction: The meanings of symptom clusters for patients with chronic obstructive pulmonary disease: A cross-sectional qualitative study

Fei Fei[1,2]*, Richard J. Siegert[3], Xiaohan Zhang[2], Jonathan Koffman[4]

**1** School of Medical and Health Engineering, Changzhou University, Changzhou, Jiangsu, China, **2** Cicely Saunders Institute, Florence Nightingale Faculty of Nursing, Midwifery and Palliative Care, King's College London, London, United Kingdom, **3** Psychology, Auckland University of Technology, North Shore, Auckland, New Zealand, **4** Wolfson Palliative Care Research Centre, Hull York Medical School, University of Hull, United Kingdom

* feifei1@cczu.edu.cn

## Abstract

### Purpose

Symptoms in chronic obstructive pulmonary disease (COPD) commonly co-occur as 'symptom clusters', yet little is known about how patients interpret and make sense of these experiences. This study aimed to explore the meanings patients with COPD attribute to symptom clusters, their impact on health-related quality of life, and the strategies used to manage them.

### Methods

A qualitative study using semi-structured interviews was conducted with a purposive sample of 30 patients with COPD recruited from a university teaching hospital in China. Data were analysed using the Framework approach, informed by the Theory of Unpleasant Symptoms and the concept of biographical disruption.

### Results

Three interrelated themes were identified. First, participants understood symptom clusters as dynamic, interacting experiences, often organised around "trigger symptoms" (e.g., cough precipitating breathlessness and fatigue), rather than as isolated symptoms. Second, these clusters disrupted multiple dimensions of everyday life, including physical functioning, social participation, and sense of self. Third, participants actively negotiated symptom burden through a range of adaptive strategies, prioritising symptoms based on their perceived meaning and impact rather than clinical severity. These findings highlight how symptom experiences are shaped by both physiological interconnections and personal interpretations.

**Data availability statement:** The qualitative data generated and analysed during this study are not publicly available due to the sensitive nature of the interview content and the risk of participant identification. De-identified excerpts supporting the findings are included within the manuscript. Additional materials, including the coding framework and thematic framework matrix, may be made available on reasonable request. Requests for access should be directed to the Cicely Saunders Institute (King's College London) via [palliativecare@kcl.ac.uk] and will be subject to review to ensure that the proposed use is consistent with the original ethical approvals and participant consent. Applicants will be required to provide a clear research proposal and agree to data use conditions designed to protect participant confidentiality.

**Funding:** China Scholarship Council (grant numbers 201806170005) We also confirm that the funder (China Scholarship Council, grant number 201806170005) had no role in study design, data collection and analysis, decision to publish, or preparation of the manuscript.

**Competing interests:** The authors have declared that no competing interests exist.

## Conclusions

This study provides novel insight into how people with COPD experience symptom clusters as interconnected, meaning-laden phenomena. Recognising the role of "trigger symptoms" and patients' subjective prioritisation of symptoms has important implications for person-centred assessment and supports the development of targeted, mechanism-informed approaches to symptom management in COPD.

*I can't walk long distances. I can't go out and spend time with my friends. This is a big problem in my life.* (69-year-old male, GOLD stage III)

*Other people can run around. They visit relatives or friends during the holidays. I can't. I cough all day long and can only stay at home.* (63-year-old female, GOLD stage IV)

## 1 Introduction

Chronic obstructive pulmonary disease (COPD) is projected to affect approximately 600 million people globally by 2050 [1]. Globally, the prevalence of COPD is rising due to tobacco smoking and long-term exposure to toxic fumes or dust [2,3]. The burden on healthcare systems and individuals is considerable. Studies have reported large variations in the prevalence and incidence of COPD across different countries [4]. For example, in the USA, the prevalence of COPD is estimated at 2,527.7 per 100,000 in high sociodemographic index regions [5–6]; Europe, an estimated 40.1 million cases were reported in 2021 with projections showing increase from 36 million (2020) to 49 million cases (2050) [7]; Austria, an estimated 1,047,150 people over age 40 were affected in 2005 with an 88.5% underdiagnosis rate [8] and China, an estimated 99.9 million people aged 20 years or older have COPD, contributing to the global total of 213.4 million prevalent cases in 2021 [9].

People living with COPD often experience multiple concurrent symptoms that tend to co-occur and interact with one another [10,11]. This phenomenon is referred to as a 'symptom cluster'. The concept of 'symptom clusters' was first introduced to oncology symptom science by Dodd and colleagues in 2001, who proposed that symptoms occur in groups of three or more concurrent symptoms that are related to one another [12]. The most recent expert panel defines a symptom cluster as a stable group of two or more concurrent symptoms with temporal characteristics, which are independent of other clusters and may share underlying mechanisms and/or outcomes [13]. Research on symptom clusters originated in nursing science, with seminal work by Dodd et al., and has since evolved across multiple disciplines. This body of work has generated evidence on cluster composition, prognostic significance for clinical outcomes, and their potential utility in identifying targets for intervention [14].

This body of work spans various contexts, including individuals receiving chemotherapy [15], those living with advanced cancer [12], heart failure [16], head and neck cancer [17], and lung cancer [18]. As with other illnesses, the presence of multiple concurrent symptoms in people living with COPD is often related to poor quality of life, where their care for those living with them is associated with substantial healthcare resource use [19–21].

Given the use of network analysis to identify symptom clusters, it may be unnecessary to assume that these clusters are independent of each other, as symptoms often exhibit residual correlations after being grouped into clusters [14]. Inconsistent findings on the composition of symptom clusters among people with COPD were reported [22–24], which may be a consequence of varying study designs, the absence of standardised measurement tools and statistical methods [25] and especially inadequate evidence of temporal characteristics of the symptoms within a cluster [13]. Srirat et al. [26] reported a 75% to 100% similarity in symptom clusters across both severity and distress dimensions. Additionally, four core symptoms remained more than 75% stable between two time points with a four-week interval among people with COPD. Since four weeks may not fully capture the changes in the progression of COPD, it is necessary to further explore whether symptom clusters are "stable" (i.e., change over time and/or across symptom dimensions) and whether specific symptoms are "consistent" within a cluster (i.e., remain the same over time and/or across symptom dimensions) [14].

To date, most studies investigating symptom clusters in COPD have adopted quantitative designs, identifying symptom groupings through a range of questionnaires and statistical methods [25]. While such work has advanced knowledge in the field, its findings are predominantly shaped by statistically derived relationships rather than insights grounded in clinical practice or patients' own narratives [27,28]. Qualitative research offers an alternative approach to exploring symptom clusters, providing rich, contextually embedded insights into how patients experience multiple concurrent symptoms and perceive associations among them through experiential data [29]. Despite the growing number of studies that have explored the meanings attributed and experiences to symptom clusters present in other diseases [30–32] and how people cope with them, few studies have been undertaken among those with COPD. Therefore, our research questions were developed deductively from the identified gaps in the existing symptom cluster literature [25,30–32] and inductively from established conceptual frameworks emphasizing symptom composition, patient impact, and coping strategies [13,14]. Consistent with exploratory qualitative methodology, the questions were formulated to be open-ended to elicit rich experiential data without imposing predetermined hypotheses [29].

This study, therefore, attempts to address the following questions:

1. What are the symptoms experienced concurrently by patients living with COPD?

2. What narratives do patients adopt to describe symptom clusters associated with COPD?

3. In what ways do symptom clusters associated with COPD impact their lives?

4. How do patients with COPD adapt and cope with the symptom clusters associated with their COPD?

The study incorporates two theoretical approaches specifically 'Theory of Unpleasant Symptoms [33]' and 'biographical disruption' [34]. The Theory of Unpleasant Symptoms (TOUS) represents a holistic 'middle-range' theory that integrates a structured and comprehensive mechanism to inform researchers from different perspectives on how to conduct symptom research where a variety of symptoms are present [33]. Specifically, the TOUS assumes that symptoms can occur alone or with multiple other symptoms frequently resulting in multiple relationships between and among symptoms [35]. In the TOUS, symptoms are recognized as multidimensional, encompassing timing, intensity, distress, and quality. Timing refers to the duration or frequency of a symptom. Intensity and distress denote the severity of a symptom and the extent to which it bothers the patient, both of which can be quantified through measurements. Quality pertains to how a symptom is perceived by the patient [33]. Physiological, psychological and situational factors may influence these symptoms [36]. Furthermore, the TOUS posits that symptoms that group into distinct clusters will have an impact on health-related quality of life [37]. This theory helped us to appreciate the experiential nature of symptoms and led to our interest in exploring how people with COPD attribute meaning (i.e., the quality dimension) to their group(s) of symptoms and how these symptoms impact their health-related quality of life.

Bury's research suggests that experiencing chronic illness (e.g., COPD) can be conceptualized as a 'critical event' that disrupts an individual's life trajectory or "biographical disruption" [34,38]. William#39;s extension of biographical disruption

suggests that being diagnosed with an illness, for example, COPD inevitably impacts an individual's sense of self-identity and quality of life [39]. Gysels and Higginson observed that people's experiences of time were disrupted by COPD [40]. Jowsey et al. [41] also reported COPD was experienced as a disruptive event to individuals' daily rhythms and their over-all biography. Within the interpretive sociology of chronic illness, Bury's [38] work explicitly sought to pay more attention to the actions that individuals actively seek out to counter the impact of their illnesses [38]. In addition, Bury's concept outlined that 'strategy' or 'strategic management' is the means that individuals draw upon to minimise the negative impact of this biographical disruption [42]. Specifically, 'strategy' represents 'the actions people take, or what people do in the face of illness' [38]. In this study, the TOUS will be operationalized to identify and characterize the multidimensional nature of symptom clusters experienced by participants, while the 'strategy mechanism' of biographical disruption will provide an interpretive lens for understanding how participants actively respond to, manage, and mitigate the impact of these clusters on their daily lives and self-identity [33,38]. Therefore, we wished to understand how patients living with COPD actively adopted the 'strategy' mechanism to manage their disrupted biographies as a consequence of the symptom clusters they experienced.

## 2 Methods

### 2.1 Design

A descriptive qualitative study presents a comprehensive summary of phenomena, particularly healthcare phenomena [43] The Consolidated Criteria for Reporting Qualitative Research (COREQ) checklist [44] was adopted to improve the transparency of the present research (See S1 File). Overview of methodological approaches and their role in the study, a file (See S2 File) provides a clear and concise overview of how the study design, data collection methods, analytical approach, theoretical frameworks, and quality criteria fit together, thereby improving clarity and readability.

### 2.2 Ethics

All participants provided written informed consent prior to participating. Research ethics approvals for this study were obtained from researcher's institutional review board (approval number: HR-18/19–13608), and also obtained ethical approvals from the Second People's Hospital of Changzhou Ethics Committee (approval number: [2019] KY063−01), the Affiliated Huai'an No.1 People#39;s Hospital of Nanjing Medical University Ethics Committee (approval number: KY-P-2019-048-01) and the Affiliated Huai'an Hospital of Xuzhou Medical University Ethics Committee (approval number: HEYLL201932).

### 2.3 Setting and participants

COPD patients, now referred to as participants were recruited from the respiratory department of a university teaching hospital located in XX. Study inclusion criteria included adult patients who were 18 years of age or older, diagnosed with COPD by clinicians with pulmonary function tests of forced expiratory volume in one second (FEV1)/forced vital capacity (FVC) ≤ 70%; and able to provide written informed consent to participate were recruited for this study. Patients with mental illness, language dysfunction and other serious physical illnesses were excluded.

### 2.4 Recruitment and sampling

Eligibility was confirmed through brief clinical screening by the treating clinician, who used professional judgement to ensure that participants had the capacity to understand the study information, provide informed consent, and engage meaningfully in the interview; no formal cognitive assessment tool was used. Potential participants were provided with a copy of the information sheet and informed that the decision they made would not affect the care they received currently or in the future. The researcher prepared a response form in a sealed envelope in advance. The form asked potential

participants to indicate whether they wished to be contacted by the research fellow (RF) by selecting one of two options: (i) 'I am happy to be contacted by the RF'; or (ii) 'I am not happy to be contacted by the RF'. Space was provided for participants to include their name if they consented to be contacted. Potential participants were given at least 48 hours to consider participation. They then completed the form, placed it in the sealed envelope, and returned it to the clinician. The researcher attended the clinic regularly to collect the sealed envelopes. Individuals who indicated willingness to be contacted were subsequently approached and provided with further information about the study. A mutually convenient time was arranged to meet and conduct a face-to-face interview. Prior to the interview, the researcher re-explained the study and addressed any questions or concerns. Written informed consent was obtained from all participants before participation. Participants also provided consent for the research team to access relevant clinical records to obtain disease-related information for the study. Written informed consent was obtained from all participants prior to interview.

Purposive sampling was used to select information-rich cases across a range of characteristics, including age, gender, educational level, income, family status, payment method for hospitalisation, employment status, and disease severity (GOLD stage). This strategy was operationalised through iterative sampling from the pool of consenting participants to ensure diversity across these criteria, with active recruitment of under-represented subgroups until thematic saturation was achieved [45].

## 2.5 Data collection

Face-to-face, semi-structured interviews were conducted between November 14, 2019, and April 20, 2020, by the first author, who had no previous relationship with study participants. However, The first author possessed the same cultural language as the participants and understood important cultural cues and dynamics within conversations and dialogues. Written informed consent was obtained from all participants prior to participation. This included consent to take part in the study, to audio-record the interviews, and to use anonymised data for research purposes. Verbal consent was additionally confirmed at the start of each interview to ensure participants' ongoing willingness to participate and later transcribed as part of the interview transcripts, ensuring proper documentation. The development of our interview topic guide (see Table 1) was guided by the literature on symptom clusters theories, study objectives and subsequently piloted and refined [46] to improve its operation in the field. Clarifications and probes were used during the interviews to ensure adequate details were captured and incorporated into the final version of the interview topic guide. All interviews were audio recorded and handwritten field notes were taken to account for interview surroundings and the interviewees' non-verbal cues or expressions. The interviews varied in length between 24 and 55 minutes and were largely conducted in the afternoon when study participants had completed their treatment for the day.

## 2.6 Data analysis

The analysis proceeded through two consecutive phases. The first phase employed the Framework approach [47] to organise the data, which then enabled the second phase of interpretation. Data organisation was initiated by F.F. while interviews were still ongoing and involved thorough familiarisation with the material. All interviews were audio-recorded and transcribed verbatim. To ensure accuracy, transcripts were checked against the original audio recordings by the research team. Where interviews or excerpts were translated into English, translation was undertaken by bilingual members of the research team. Translated excerpts were reviewed to ensure that the original meaning and context were preserved. A thematic framework was developed inductively from the interview content and deductively informed by the theoretical frameworks underpinning the study. The analysis was informed by two complementary theoretical frameworks: the TOUS and the concept of biographical disruption. TOUS provided a structured lens to identify and characterise the multidimensional nature and interrelationships of symptoms within clusters. In parallel, the 'strategy mechanism' of biographical disruption informed the interpretive analysis by focusing on how participants understood, responded to, and managed the impact of these symptom clusters on their daily lives and sense of self. The combined use of these frameworks enabled both

**Table 1. Semi-structured interview topic guide.**

| Topics | Questions | prompts |
|---|---|---|
| Questions about symptoms experience | Tell me in your own words about your illness, and how you came to being here today? | •How have you generally been?<br>•Please tell me about things bother/trouble you most about your illness (e.g., pain, breathlessness), can we refer to them? |
| | Tell me about what bothers or troubles you about your illness at the moment. From here onwards we can refer to them as 'symptom' you are experiencing. | •Example: symptoms could include breathlessness, fatigue, anorexia, pain, depression, anxiety, cough, daytime sleepiness and insomnia, dry mouth and sexual dysfunction<br>•How would you describe your whole symptom experience?<br>•How would you describe your current experiences of symptoms?<br>•How severe and frequent are your symptoms and how do they differ? And how long did they last? |
| | Tell me do any of these symptoms happen at the same time? Which ones? | •Tell me in what ways they affect each other. Are some more severe than others?<br>•Has managing one symptom influenced others?<br>•Has one symptom improved/ made worse another? What was it? How did it help? Have they together interfered with or influenced your illness? |
| | If they do, tell me in what ways these symptoms individually or importantly together make you feel in yourself? | •Do you worry about having these symptoms?<br>•How bad is it? How do you react to it? (e.g., hopelessness)<br>•What's happening for you when you feel…?<br>•What are the most difficult aspects of experiencing [symptoms]? |
| | Tell me if any of these symptoms individually you have talked about, either individually or together have changed over time? | •Which ones? How did they change?<br>•Has the location of these symptoms changed?<br>•Have they changed in type? Severity?<br>•Have your symptoms reappeared after treatment ended? |
| Questions about the impact of symptoms occurring at the same time | When they occur together tell me in what ways this effect your everyday life/ activities. | •Have they affected your life, interests, well-being, self-esteem, relationships... |
| Questions about symptoms coping strategies | Tell me in what ways you are coping with or living with all these symptoms when they occur together? | •E.g. is it because you're tired, you're not coping with pain?<br><br>•Tell you in your own words what helps you and how helps you. |
| | When these symptoms occur together in what ways have you tried do to reduce their impact on your life? Have you found any ways which help you? Why do you think this might be? | •What are strategies which help you?<br><br>•What has made your [symptoms] better and worse?<br><br>•Could you describe how you've tried to deal with multiple symptoms?<br><br>•Is there anything physical that helps you cope? |

descriptive and interpretive depth in the analysis. The use of TOUS and the concept of biographical disruption contributed to analytical rigour by providing theoretically informed frameworks that supported consistent and conceptually grounded interpretation

To enhance the credibility of the findings, we employed triangulation. First, data source triangulation was used by comparing participants' interview accounts with relevant clinical information (e.g., GOLD stage, spirometry results) to contextualise symptom experiences. Second, analyst triangulation was undertaken, whereby two researchers independently coded the data and discussed emerging interpretations, with discrepancies resolved through discussion with a third researcher. This process strengthened the rigour and trustworthiness of the analysis. Objective clinical data (e.g., pulmonary function test results, GOLD classification) were used solely to confirm eligibility and to provide clinical context for the sample. The qualitative analysis focused on participants' subjective experiences of symptoms as described in interviews. Clinical signs and findings from physical examination were not analysed as data within this study but formed part of the broader clinical background in which participants' experiences were situated. The provisional thematic framework was developed through iterative deliberation among the research team (X.X., X.X., and X.X.). Its robustness was subsequently tested and refined via independent coding of ten transcripts, a process facilitated by NVivo 12 software. Preliminary findings from the first phase of interviews were subjected to collective discussion and consensus among the team. One researcher (X.X.) subsequently charted the data from each interview into a matrix, organizing them according to emergent themes and corresponding sub-themes. This matrix enabled dual-level analysis, permitting examination at both the case level (within and across individual participants and participant groups) and the thematic level (across and within themes). We have made some use of numerical and verbal counting as this can help to clarify patterns emerging from the data while recognizing that the main emphasis is to identify meanings and conceptual categories [48]. Analytical rigour was underpinned by a realist approach [49,50], drawing on established quality criteria for qualitative research [44,51]. The specific criteria and the corresponding strategies implemented are summarised in Table 2.

## 3 Results

### 3.1 Sample characteristics

A total of 30 COPD patients participated in this study. Nine people declined – refer to the flow chart in Fig 1 for their reasons. The detailed demographic and clinical characteristics of the participants who participated in this study are summarised in Table 3.

### 3.2 Main findings

The analysis of the interview transcripts identified three substantive themes relating to the participants' accounts of their symptom experiences and coping strategies for multiple concurrent symptoms (Fig 2).

**3.2.1 Theme one: Meanings of symptom clusters.** The theme 'meanings of symptom clusters' specifically refers to how participants cognitively attributed meaning to their complex experience of multiple concurrent COPD-related symptoms. Participants commonly reported symptom clusters associated with dyspnoea. Regardless of the specific symptom clusters reported, patients always used the term "trigger" to describe the relationship between symptoms, such as coughing triggering dyspnoea. In this study, 'trigger' symptoms refer to symptoms that appear to initiate or exacerbate a cascade of interconnected symptoms. Additionally, patients' experiences of symptoms varied throughout the day. Daytime activities might trigger a "cough-dyspnoea-fatigue" symptom cluster, while at night, they might experience either an "anxiety-irritability-sadness-fear-dyspnoea" cluster or a "cough-dyspnoea-sleep disturbance" cluster. This variation reflects the multidimensional experience of symptom clusters at different times and their dynamic impact on patients' quality of life.

**3.2.1.1 Typologies and Interrelationships of Symptoms:** Participants described six main symptoms associated with their COPD including dyspnoea, cough (with or without phlegm), fatigue, mental problems (i.e., feeling anxious, irritable, sad, or fearful), dry mouth and difficulty sleeping. Among these symptoms, dyspnoea was considered the most distressing. From these narratives, four symptom clusters were observed to be present.

**Table 2. Quality criteria selected for ensuring rigorous qualitative analysis [49,50].**

| Quality criteria | How it was achieved |
|---|---|
| Rich rigour – analysis used appropriate sample, context and data-driven by theory | A purposive sample of 30 individuals with COPD was recruited. Semi-structured interviews enabled participants to narrate their experiences without restriction. We drew on two theoretical frameworks namely the TOUS and biographical disruption to facilitate analysis of the primary data. TOUS provided a structured lens for identifying and characterizing the multidimensional nature of symptom clusters, while the 'strategy mechanism' of biographical disruption enabled interpretation of how participants actively responded to and managed these clusters. Theoretical integration – Use of TOUS and biographical disruption frameworks – Provided a structured and conceptually grounded lens for data interpretation, enhancing analytical depth, consistency, and transparency in linking data to themes. |
| Credibility and authenticity –thick descriptions and detailed findings have been provided to support inferences | The qualitative dataset generated rich, contextualised accounts that capture the complexity inherent in symptom-cluster experiences among individuals with COPD. The findings reflect participants' lived experiences as they were conveyed, with illustrative quotations drawn from a diverse range of interview transcripts to substantiate the interpretations. |
| Criticality – a detailed account of how researchers critically appraised their findings | The analytical process was documented in full. Two researchers (F.F. and X.Z.) independently coded the data and subsequently compared their findings, enabling critical and alternative readings of the material. They engaged in regular, reflective dialogue to scrutinise assumptions and to consider how their respective backgrounds might influence interpretation. Divergent interpretations were discussed in depth until consensus was reached, with a third researcher (J.K.) consulted as necessary. To ensure fidelity to the original language and cultural context, F.F. and X.Z. analysed the original language through independent coding before a consensus was achieved on the categories and themes in English, which ensured the accuracy of the participants' intended meanings within the given socio-cultural context |
| Attention to contradictory or non-confirmatory data | Throughout the analysis, F.F., X.Z., and J.K. remained attentive to data that challenged or diverged from the emerging themes. Such instances were integrated into the subsequent development of the themes and are reflected in the reporting of findings. |
| Fidelity or meaningful coherence – analysis achieves its intended goals by using appropriate methods | To address the research questions, we maintained a coherent 'thread' across the study, from the recruitment strategy and topic guide through the interview approach, analytic plan, and reporting of findings, ensuring consistency with our interpretive aims. |

The 'cough-dyspnoea-fatigue' cluster was the most frequently reported. Participants attributed this to irritation from eating overheated food or inhaling cold air among, other factors. They also described 'triggers' for their cough (with or without phlegm), dyspnoea and fatigue, typified by the following examples:

*I have always had inflammation (points at her lungs). I always have phlegm! When I eat. something hot, this (points to her trachea) becomes itchy and once it becomes itchy, I have phlegm. When I cough and produce phlegm, I gasp harder and don't feel like doing anything.* (90-year-old male, GOLD stage II)

*When I cough, I gasp; when I gasp, I do not dare to move, because when I move, I gasp even harder. When I cough too much, I lose energy. In fact, even when I have done nothing, I feel I have done many things and feel very tired.* (63-year-old female, GOLD stage IV)

The 'anxiety-irritability-sadness-fear-dyspnoea' cluster was present among six participants suggesting a relationship between physiological and psychological symptoms. Feelings of anxiety, irritability, sadness, or fear are known to trigger dyspnoea. Additionally, dyspnoea triggered negative emotions among participants. They described experiencing a vicious cycle of physiological and psychological symptoms illustrated by the following accounts:

*I feel anxious night after night. When I feel anxious, I gasp more. I can't tell. others about it because I worry that they will laugh at me.* (77-year-old female, GOLD stage II)

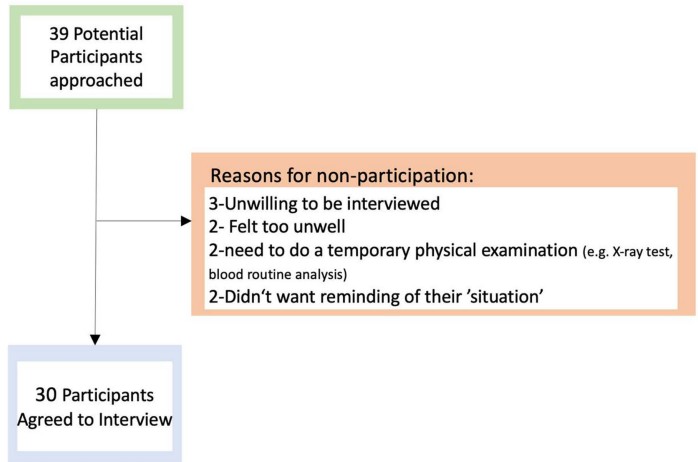

**Fig1. Flowchart of participants recruitment and reasons for non-participation.**

*At night, when people are asleep I remain awake. I get more anxious when I cannot fall. asleep and I gasp for breath even harder when I'm feeling anxious.* (63-year-old female, GOLD stage IV)

*It's better not to get angry. I know this. However, I can't control my temper. When I get angry, I need to gasp. When I am angry, I feel very uncomfortable. My chest feels constricted, and I can't breathe.* (75-year-old male, GOLD stage III)

Participants described the 'cough-dyspnoea-sleep disturbance' cluster most frequently as occurring at night. Some reported that when they coughed at night this triggered dyspnoea and as a direct consequence made sleeping especially difficult:

*I cough badly. I have phlegm, but it doesn't come out. I cough and gasp and cannot sleep well at night.* (70-year-old male, GOLD stage III)

*I cough a lot at night. I don't cough that much during the day. At night, when I begin to cough, I gasp. I cannot fall asleep. I just lie there and feel extremely sleepy.* (63-year-old female, GOLD stage IV)

The 'dry mouth-cough-dyspnoea' cluster reported by participants was distinctive but when compared to others represented a less dramatic manifestation of symptoms among those interviewed. However, as indicated by some participants, a dry mouth often triggered cough which, in turn, caused them to experience dyspnoea as indicated by the following two participants:

*When I am thirsty, my throat feels dry which causes me to cough which further exacerbates me to gasp.* (64-year-old male, GOLD stage III)

*I feel thirsty, especially at night. I always cough, produce phlegm and feel thirsty. I have had Charting facilitated using NVivo 12 software for these symptoms for a long time. I cough, and then I gasp. I gasp so hard that I can't even go to the bathroom.* (73-year-old male, GOLD stage II)

**3.2.1.2 Variable nature in concurrent symptoms:** In this study, we observed that symptom clusters associated with COPD varied among participants, influenced by their different demographic, clinical, and individual

**Table 3. Summary of the demographic and clinical characteristics of the patients (N=30).**

|  | Patients (n=30) |
| --- | --- |
| Mean age in years (range) | 74(63-90) |
| Gender |  |
| Male | 23 |
| Female | 7 |
| Gold stage |  |
| I | 4 |
| II | 2 |
| III | 7 |
| IV | 17 |
| Family Status |  |
| Married, living with a spouse | 18 |
| Married, living with other family | 9 |
| Divorced, living alone | 2 |
| Widow, living with other family | 1 |
| Educational level |  |
| Illiteracy/Uneducated | 3 |
| Primary school | 9 |
| High school | 14 |
| ≥Vocational school | 4 |
| Hospitalization payment |  |
| Publicly funded free medical care | 4 |
| Medical insurance | 25 |
| Self-funded | 1 |
| Monthly income per person (Chinese) |  |
| <500 | 4 |
| 500-1000 | 4 |
| 1000-3000 | 4 |
| >3000 | 18 |
| Employment status |  |
| Retired | 24 |
| Farming | 6 |
| Time since diagnosed (Years) |  |
| 1-10 | 16 |
| 11-20 | 12 |
| 21-30 | 1 |
| >30 | 1 |
| Oxygen home therapy |  |
| Yes | 21 |
| No | 9 |
| Smoking history |  |
| Ex-smoker | 23 |
| Current smoker | 2 |
| Never smoked | 5 |

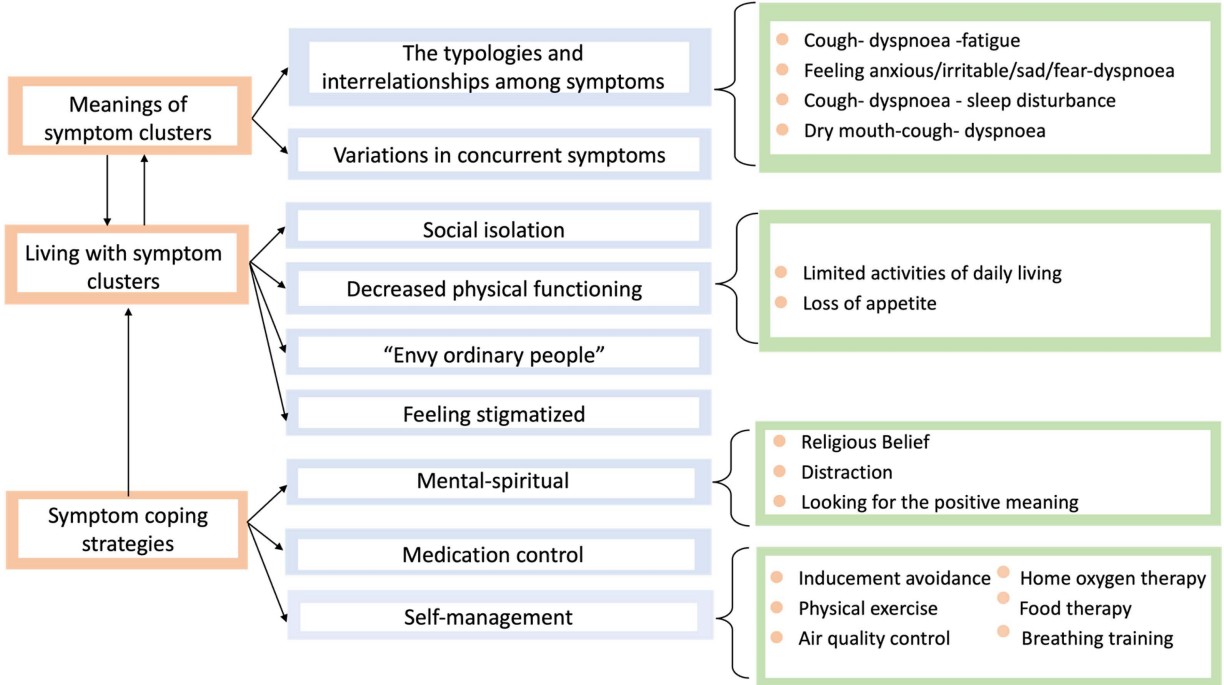

**Fig 2. Study themes and subthemes.**

characteristics. Additionally, symptom clusters appeared to change in nature throughout the day. Some participants reported experiencing a "cough-dyspnoea-fatigue" cluster as a result of engaging in certain daily activities, which aggravated their dyspnoea symptoms and fatigue. In contrast, at night, participants described experiencing either an "anxiety-irritability-sadness-fear-dyspnoea" cluster or a "cough-dyspnoea-sleep disturbance" cluster. They believed that their isolation from others, who were comfortably asleep, made them emotionally sensitive and vulnerable to cough and its associated symptoms. A participant evocatively described the associated changes as follows:

> At night, when people are asleep, I remain awake. I get more anxious when I can't fall asleep, and I gasp harder when I am anxious. I cough a lot at night. I don't cough that much during the day. At night, once I begin to cough, I gasp for breath. I cannot fall asleep. I just lie there…. (63-year-old female, GOLD stage IV)

> The more I cough, the more I gasp. The more I gasp, the more tired I feel. I get very tired, and I don't even want to eat. I feel anxious night after night. When I feel anxious, I seem to gasp more. I can't tell others about it because I worry that they will laugh at me. The coughing is the most difficult bit to bear. I cough every night and every day. When I cough, I pant, and find it difficult to sleep because I cough when I sleep. (77-year-old female, GOLD stage II)

**3.2.2 Theme two: Living with symptom clusters.** Participants' perceived recognition of symptom clusters influenced their health-related quality of life. Health-related quality of life, as a health-promoting outcome, refers to the degree to which individuals perceive themselves as healthy, comfortable, and able to participate in or enjoy life events [52]. However, for people living with COPD, this perception is compromised in several ways by the presence of symptom clusters.

**3.2.2.1 Decreased physical functioning:** All of the participants interviewed reported experiencing a profound decrease in their physical functioning. Although they were unable to understand the aetiology of their decreased physical functioning, they viewed their impaired pulmonary functioning (not described in this technical manner) as the main reason for their decreased endurance and strength. With the aggravation of their COPD, participants reported they progressively lost their ability to perform activities of daily life that were amplified by the presence of symptom clusters. This was most pronounced during acute exacerbations of their COPD. Participants stated they experienced a rapid decline in lung function, and some could not do anything but lie in bed with their breathing assisted by a ventilator. This is illustrated by the following participant who stated,

*I am seriously ill! I can't even walk. I can't even move. I can't go home. In the past, I couldn't even eat. When I cough, I gasp hard, and then I can't move. I can walk on a flat road for approximately 200–300 metres. However, I can't climb the stairs. I don't have the strength........When I cough a lot, I can't even stand up and pee. I can't stand up. When I have to prepare food, I have difficulty breathing. When I cook rice and noodles, I have difficulty breathing. In addition, I even have difficulty taking a shower. Sometimes, I need to inhale oxygen when I am getting dressed.* (78-year-old male, GOLD stage III)

The combined presence of dyspnoea and endless coughing prevented several participants from eating easily. Moreover, they reported physical sensations of a feeling 'swollen' and having 'stiff lungs' which may have been caused by pulmonary fibrosis which was associated with stomach tension:

*I gasp when the symptoms flare up. I can't breathe properly! I don't want to eat and feel. as if my stomach is blocked. I wish my stomach felt smaller.* (64-year-old male, GOLD stage III)

*I don't want to eat anything when I gasp. Now, my lungs feel like they are inflated. They. Seem to push on my stomach which makes me feel uncomfortable.* (64-year-old female, GOLD stage III)

**3.2.2.2 Social isolation:** As a result of changes caused by the symptoms associated with their COPD participants spoke of being forced to limit or even end social interactions with family and friends. This biographically disruptive aspect of their illness led to profound isolation [38,41] and a considerable narrowing of the lives many participants previously took for granted. They reported they found this distressing which, in turn, limited the instrumental and emotional support a number stood to receive:

**3.2.2.3 "Envying ordinary people":** "Aesthetic" refers to the appreciation of and search for beauty, balance, and form [53]. This subtheme was present with one participant who expressed a view that she could no longer wear her favourite dress because she believed she would experience an acute exacerbation of the concurrent symptoms of cough, gasp and fatigue associated with her COPD as a consequence of catching a cold:

*I envy ordinary people. They can wear skirts but I can't. I would catch a cold if I wore a skirt.* (63-year-old female, GOLD stage IV)

**3.2.2.4 Feeling stigmatised:** Stigma has been defined as an "attribute that is deeply discrediting" or a "negative trait in a person" [54]. Both suggest an individual or the wider group perceives a situation of that individual being disqualified from full social acceptance. The combination of symptoms including severe dyspnoea, ceaseless cough and phlegm resulting from COPD placed some participants in a position where they felt embarrassed. Moreover, they perceived bystanders, for example, family or friends who witnessed these symptoms, may have felt uncomfortable on their behalf. A number complained that they often felt misunderstood by others who were not aware of COPD or the symptoms associated. They felt concerned and stressed about experiencing prejudice from others illustrated by the following two participants who stated.

*My son is not worried about being "infected" but my daughter-in-law is. When I want to take care of her children, I have to wear a face mask or else she thinks the kids will get infected. I feel so sad.* (77-year-old female, GOLD stage II)

*In the past, I played with kids in my neighbourhood every day. Now I can't. People would gossip about me if I played with them while sick.* (74-year-old male, GOLD stage III)

**3.2.3  Theme three: Symptom coping strategies.**  After being diagnosed with COPD, participants were confronted with many new challenges in adapting to their new situations, often with varying degrees of success. The perception of symptoms and symptom clusters differed among patients, leading to the adoption of diverse self-management strategies. Despite the perceived upheaval of their COPD and the multiple concurrent symptoms they experienced, participants cultivated a range of strategies to help them cope. This theme refers to participants' efforts to manage the symptom clusters they experienced to maintain their quality of life. Their coping mechanisms were tailored to their individual perceptions and experiences of symptoms, highlighting the variability in self-management approaches among those living with COPD.

**3.2.3.1 Mental-spiritual strategy:** Religion provided some female participants with a framework to comprehend their COPD and symptom clusters as being ordered and explicable rather than a disordered and inexplicable life event. Specifically, prayer was perceived as possessing healing powers and was considered to be an essential strategy that helped optimise the way they coped with living with distressful concurrent symptoms and alleviation of symptom-related distress. Religious observations became a daily routine practice for these women. Two participants, who stated they were of Christian faith, reported that the Bible and Christian hymns helped them stabilise their combination of being breathless and anxious at the same time which led to them feeling more at peace and comfortable. Moreover, the strength resulting from a strong belief in God represented the physical and almost tangible capacity to deal with the relentless impact of their COPD symptoms.

*I am a Christian, and I feel that God can give me comfort. If I did not have this faith, I would have been much sicker.* (72-year-old female, GOLD stage III)

*I believe in Jesus. When I am out of breath, I listen to Bible readings on the radio or read the Bible myself. I sing hymns to myself when I feel good.* (80-year-old female, GOLD stage III)

Some participants attempted to divert their attention from worrying about their breathlessness and anxiety by engaging in distracting, recreational activities:

*If I have insomnia I get up and play Tetris on my phone to help regulate my sleep.* (72-year-old female, GOLD stage III)

*I brought playing cards to the hospital, and when I cannot fall asleep, I just play poker by myself in bed.* (70-year-old male, GOLD stage III)

*I also like to read the news. It is good for my spirit. I try to keep myself busy; in a positive. spirit is very important.* (88-year-old male, GOLD stage III)

In several instances, participants sought to identify positive interpretations of the distressing symptom combinations they were experiencing to support themselves in their struggle against their illness.

*I wear this ring for peace of mind and I also think it looks good on me. Reading good books also has a calming effect on my mood.* (72-year-old male, GOLD stage III)

**3.2.3.2 Medication control:** Medications were considered to be vital in controlling COPD symptoms and delaying disease progression. All participants displayed good concordance with their medication treatments. Most participants were well informed of the effects and doses of their drug regimens to control some of the symptoms associated with their COPD typified by the following comments:

*I usually use salbutamol where two or three sprays give me relief.* (73-year-old male, GOLD stage III)

*When coughing or wheezing I will take anti-inflammatory drugs and cough and asthma medicine.* (65-year-old male, GOLD stage III)

**3.2.3.3 Self-management strategy:** Some participants developed symptom self-management strategies to relieve their COPD-related symptom cluster burden. The participants reported that catching a cold or inhaling smoke resulted in acute exacerbations of the distressing concurrent symptoms. They consequently attempted to take measures to minimise the possibility of being exposed to these triggers:

*My main symptom is gasping. Once I move, I gasp. I cannot take care of myself now. On the other hand, phlegm is also a serious problem. But once I have the disease, I will produce phlegm. It feels terrible. It's like a living death. If the phlegm is not coughed out, I will cough harder, and gasp harder. Therefore, it is necessary to keep warm and not get cold! I have a physical therapy bed at home that can generate heat. Basically, I stopped going out from the end of October to May of the following year, so I do not catch a cold! The main thing is not to catch a cold since a cold will come with other problems!* (74-year old male, GOLD stage IV)

*When I have phlegm in my throat I want to cough. But the coughing makes me feel like gasping. After the cough and after I produced the phlegm I feel better and my chest becomes comfortable again.* (69-year-old male, GOLD stage III)

Almost all the participants received long-term oxygen treatment at home, and severe cases of COPD were treated with non-invasive ventilation. Participants used home oxygen therapy or non-invasive ventilation according to their own needs on flexible schedules, which had a positive impact on the symptom clusters they experienced:

*I am on a ventilator when I sleep to assist with ventilation but it also helps me expel carbon dioxide. Then my breathing becomes much better and I don't feel so tired after doing even a little exercise.* (74-year-old male, GOLD stage IV)

When in relatively 'stable' health, some participants performed physical exercises as their capacity allowed such as Qigong, Tai Chi and the 'five poles exercise'. Specifically, Qigong represents a 'mind-body' practice first developed over 5,000 years ago and is an important part of traditional Chinese medicine used to promote health, well-being and improve medical conditions [55]. Tai Chi (also referred to as Taiji) is also a traditional Chinese wellness practice. In addition, 'Qigong and Tai Chi' are close relatives having shared theoretical roots, common operational components, and similar links to the wellness and health-promoting aspects of traditional Chinese medicine [56]. All these exercises were intended to help individuals improve the distress associated with symptom clusters they experienced as illustrated by the following participant:

*I do the "lung-beat exercise", which involves beating my lung meridian which stems from the lung meridian of Chinese traditional medicine. I get up every morning at 5:30 AM to do this exercise.* (81-year-old female, GOLD stage III)

Similar to their use of physical exercise, participants also made use of food therapy-based approaches that included traditional Chinese medicine to improve their health and alleviate the effects of some of the symptom clusters they

experienced. Participants reported that eating food with lung clearing, cough relief, phlegm resolution and immune-enhancement properties three meals a day helped them with the combination of symptoms they experienced:

*I eat phlegm-relieving foods like radishes and I try not to eat spicy and irritating food. As soon as I eat spicy food, I start to cough. I have phlegm and feel breathless.* (75-year-old male, GOLD stage III)

*Every day I cook with mint, lotus seeds, silver fungus, red dates, wolfberries and all that. Silver fungus is good for the lungs. Silver fungus is good for lung clearing. I think I'm getting some relief from phlegm and difficulty breathing.* (78-year-old male, GOLD stage II)

Participants noted living in a healthy environment without air pollution as being specifically vital in controlling the concurrent symptoms of cough and difficulty breathing. Avoiding air pollution and ensuring indoor air circulation were considered critical measures:

*When I experience a cough and feel it is hard to breathe, I use a little handheld electric fan, which is enough to keep me comfortable. The natural wind is cooling.* (64-year-old male, GOLD stage III)

*I bring a mask when I am travelling in a polluted environment. Dust and smog are not good for breathing and cough.* (67-year-old male, GOLD stage II)

Participants were taught 'pursed-lip breathing' and 'diaphragmatic breathing, which have been proven to increase lung capacity, respiratory motion and peripheral oxygen saturation and reduce respiratory rates:

*The nurse has taught me how to do half-closed lip abdominal respiration, and I can do it myself. It actually helps me relieve my breathlessness and anxiety. (*72-year-old male, GOLD stage III)

*The doctors taught us to breathe deeply and practice blowing up balloons.* These exercises improve lung function. (73-year-old male, GOLD stage II)

## 4 Discussion

This qualitative descriptive study derived first-hand from patients' narratives provides a novel understanding of the symptom clusters of those living with COPD and how these concurrent symptoms impact their health-related quality of life. In addition, the findings of this study present a range of strategies adopted to better cope with the multiple, often concurrent, symptoms they experienced. The findings from this study provide new evidence that symptom clusters associated with COPD can be viewed as being highly disruptive to daily life and participants' biographies. The disruption to biography was most present concerning the many limitations participants reported about their activities of daily living. While our findings are exploratory and novel, they may, nonetheless, have value in facilitating personalised interventions for managing symptom clusters in COPD participants. The purpose of TOUS is particularly valuable in contributing to the healthcare professionals'understanding of the human experience of symptoms in their practice in these situations. Therefore, the complexity of the TOUS model recognizes healthcare professionals' knowledge and explains how actions can be associated with profound changes in the actual COPD symptom experience [57].

The data indicate that whilst the studied participants experienced symptom clusters every day, they were not always aware of the associations among the different symptoms and often focused their narratives on individual symptoms. These findings are consistent with published qualitative studies exploring experiences with multiple concurrent symptoms in those living with cancer [58,59] and those with end-stage renal disease [60]. This observation may reflect the complex,

multi-factorial nature of symptom perception [58], as participants did not regard their symptoms as discrete, objective phenomena but rather as deeply personal, meaning-laden experiences that resist straightforward articulation [61]. Furthermore, participants did not consider all symptoms to carry equivalent weight; rather, they prioritized certain symptoms over others in accordance with the meanings those symptoms held for them. Respiratory healthcare professionals involved in the management of symptom clusters in people living with COPD should therefore pay more attention to teasing out the meanings attributed to symptom experiences and explore to what extent multiple concurrent symptoms are present and in what ways they precipitate one another. Respiratory healthcare professionals have been criticised for patronising patients by ignoring their 'illness narratives' or the meanings that govern how they comprehend and accommodate their illness [62]. Instead, illness narratives should be viewed as a significant source of information in the overall process of arriving at a more complete picture of a clinical problem and helping to sort through and resolve problems associated with symptom clusters.

Our findings support other qualitative studies that observed that the symptom dimension associated with the meanings attributed to them was a stronger predictor of symptom burden than the symptom dimensions of intensity, frequency and distress [63]. Therefore, our findings provide insight into a basis for the development of patient -centred and meaning-based multiple-symptom interventions, targeting symptoms that are of the greatest subjective significance to participants' lives. Additional research is needed to substantiate this approach. Of note, four groups with multiple co-occurring symptoms were identified in participants' narratives of individual symptoms and appeared to influence each other, suggesting that these symptoms might occur in clusters.

Our findings added to the literature by noting interconnections between each symptom cluster and highlighting the existence of core 'respiratory distress' symptom pairs, including cough and dyspnoea, which occurred together and influenced each other. These findings are consistent with several published quantitative studies of participants with COPD that identified the 'respiratory-related symptom cluster' [22,24,26,64]. The findings from this study suggest four typologies of interrelationships among symptom clusters. A cough can trigger dyspnoea; persistent coughing may lead to fatigue, while reduced oxygen saturation associated with dyspnoea can further deplete energy levels, illustrating a physiological interrelationship between these symptoms. In addition, psychological problems are attributed to dyspnoea, cough and dyspnoea together with sleep problems, and dry mouth triggers cough and dyspnoea symptoms.

Our findings highlight the importance of 'trigger' symptoms of a cluster which may result in the development or exacerbation of other associated symptoms. Given the patient-centred narratives from this study that tentatively indicate their existence future research should pay more attention to identifying the nature and aetiology of 'triggering' symptoms of a symptom cluster. Unlike previous quantitative studies that had described symptom clusters in participants with COPD [25], our findings also highlight inter-individual variations in participants' concurrent symptoms.

In our study, participants perceived symptom clusters as having a negative life-depleting impact on their health-related quality of life. This closely accords with Charmaz's research where she states, *'The language of suffering severely debilitated people spoke was a language of loss"* [65]. Participants' narratives revealed that their health-related quality of life was disrupted when they learned about their disease and experienced multiple symptoms [34,38]. This marked the beginning of a 'chaotic state' in their lives [66], during which they began to realize that their quality of life had been significantly disrupted [39,67]. Specifically, the effects of multiple concurrent symptoms on physical functioning have been reported in previous quantitative studies [25,68] which showed that dyspnoea was associated with fatigue and was a strong determinant of physical performance in COPD participants [69].

Chronic illness not only disrupts the patient's physical and psychological state but also profoundly impacts their social relationships and ability to mobilise resources [38]. An interesting finding from our study was that participants' social needs were hindered by embarrassment surrounding their symptoms and the strong influence of Chinese culture. In Chinese culture, sharing information about one's unpleasant symptoms is not considered polite or culturally appropriate. Therefore, such information may become stigmatised and only shared with very trusted people [70]. An emphasis on eating is also

the norm in Chinese culture; different foods convey different spiritual meanings and affect interpersonal relationships [71]. Therefore, restrictions on or changes in dietary patterns may affect Chinese participants' social relationships and self-concept. In terms of aesthetic needs, consistent with other qualitative studies exploring the experience of long-term illness in female participants [72], our informants shared concerns about their altered appearance which limited their ability to achieve and maintain an idealised standard of feminine beauty [73].

According to Bury's theory [38], medicine should be viewed not only as a scientific system but also as a cultural one that deeply influences the patient's experience and social relationships. Healthcare professionals and caregivers should pay more attention to these biopsychosocial concerns when caring for people living with COPD. Our findings indicated that most participants had low socioeconomic status which may lead to a poorer understanding of and engagement with these participants' situations by healthcare professionals. Respiratory healthcare professionals who are in more continuous contact with patients with COPD need to focus more attention on exploring patients' feelings that are associated with their symptom-related clusters and address them with empathy and respect to satisfy their participants' individualised needs and care requirements [74].

Corroborating previous literature, our findings identified several strategies that COPD participants used to help cope with symptom clusters they experienced, namely, medication control, self-management and mental-spiritual strategies [75]. Most of our participants (90%) lived with or near their families. The family kinship system in China is different from that found in Western countries, as the Chinese family is based on a father-son-dominated kinship system, which is characterised by special features of continuity and inclusiveness [76]. Therefore, the 'family first' ideology encouraged Chinese family members to promote their family members' health and our findings showed that our participants with moderate and severe COPD tended to ask family members to send them to a hospital when their self-management strategies had failed [77]. In addition, some of the participants used Chinese herbal medicine and folk remedies. Previous studies have identified that Chinese herbal medicine can have a positive effect on the treatment of COPD [78,79]. However, some participants stated that whilst Chinese herbal medicine had a positive effect, it worked slowly and did not provide immediate relief.

In their daily lives, participants adopted many self-management strategies to cope with the disease, control their symptoms and manage acute exacerbations. The findings of this study reveal that environmental control was the main focus for our participants who used a variety of appliances or clothes (e.g., jackets and masks) to stay warm and dry. When leaving home participants avoided stimuli such as wind, extreme weather, crowds during the influenza season and stagnant air; these behaviours were also noted in a previous study [80]. Additionally, some participants engaged in 'Qigong' and ate phlegm-relieving foods (e.g., radish), mint, lotus seeds, silver fungus, red dates, and wolfberries and avoided spicy and irritating foods following traditional Chinese medicine theories of "medicine-food homology" [81]. These solutions are generally not common in Western countries. Furthermore, our participants adopted multiple coping behaviours to manage their emotions. Some participants attempted to escape their current reality and avoid thinking about the negative effects of the disease and its associated multiple symptoms. Instead, they mentioned trying to view reality through a more positive lens. In some Asian cultures, where the influence of 'Daoism' is present, Chinese people seldom express their negative emotions and usually 'accept their fate' and 'get used to symptoms' as a means to take care of themselves [82]. Our findings showed that some of our female participants managed to cope better than male patients with their everyday lives and experienced contentment from their religious beliefs. Da Silva et al. similarly found that participants with stronger religious beliefs tend to be more rigorous about their lifestyle changes and adherence to well-being activities, thus improving their quality of life. The positive effects of religious and spiritual beliefs have been observed among those living with cancer, kidney disease, mental disorders, HIV/AIDS, multiple sclerosis and cancer [83–86]. The holistic contribution of religious and spiritual care to those living with COPD requires more research to understand how respiratory healthcare professionals can be enabled to incorporate it into COPD clinical care.

## 4.1 Limitations

This study has several limitations. First, the study is cross-sectional and comprises single interviews with each participant. Therefore, it cannot provide information on changes in symptom cluster experiences and coping strategies over time. In addition, some eligible participants may have declined to take part due to practical constraints, including limited time availability or fatigue following treatment, which may have influenced participation and the range of perspectives captured. It has been suggested that prospective longitudinal recruitment is possible if a good initial relationship between interviewer and participant is developed although previous research has shown that this is often difficult because the progressive nature of disease can lead to participant attrition [87]. However, our concern is that repeated assessments could place an additional burden on participants since more than half of them (56.7%) were in the severe stage (GOLD IV) of their condition, which might raise ethical concerns. Second, our participants were Chinese and therefore the results may not be generalizable or relevant to participants of other races or ethnicities [88].

## 4.2 Relevance to clinical practice

The findings from this study have important implications for clinical practice. First, the findings suggest that participants' multiple symptom cluster experiences associated with their COPD are complex and dynamic. Assessment of symptom clusters and identification of 'trigger' symptoms should be incorporated as standard components of routine clinical assessment and physical examination for individuals with COPD. Rather than assessing symptoms in isolation, healthcare professionals should actively explore how symptoms co-occur, interact, and potentially trigger one another. This approach may enable earlier recognition of deteriorating symptom patterns and support more targeted and coordinated symptom management. A daily diary available through a 'mobile app' for recording day-to-day experiences with symptoms may be useful in clinical care to enhance the detail and validity of healthcare professionals' reports [89]. Second, it identified a broad range of multi-faceted impacts of symptom clusters on participants' health-related quality of life which could support healthcare professionals in developing holistic symptom management interventions [90]. Third, the coping mechanisms (e.g., spiritual support) and cultural perspectives identified in this study must be considered in future symptom management interventions focused on those with COPD. Fourth, the identification and management of symptom clusters and their potential 'trigger' symptoms are not confined to a single professional group or care setting. Patients with COPD interact with a wide range of healthcare professionals across the care pathway, including physicians, nurses, respiratory therapists, emergency department staff, rehabilitation teams, and community-based practitioners. These professionals, working in diverse settings such as hospitals, outpatient clinics, subacute and long-term care facilities, cardiopulmonary rehabilitation services, and home care environments, may all be the first to recognise symptom clusters and their dynamic interrelationships.Our findings therefore highlight the need for a multidisciplinary approach, in which all healthcare professionals are supported to recognise, assess, and respond to symptom clusters and 'trigger' symptoms as part of routine care. This includes ensuring that education, training, and clinical systems facilitate early identification and coordinated management across settings.

The multiple cognitive meanings attributed to symptom clusters, their varied and often significant impact and the different coping strategies patients reported closely align with the need to prioritise multi-disciplinary teamwork, coordinated care, and continuity of care to effectively manage multiple concurrent symptoms is vital. Moreover, we believe that various aspects of the health system are required to deal effectively with the complexity of concurrent symptoms. This requires a coordinated approach (see Fig 3). For this study, Fig 3 presents a model of how this approach should focus on the individual, interpersonal, system-based level, organisational, and time-based considerations present in this study. Specifically, these considerations should include how participants must be enabled and empowered to understand their symptom clusters and how to talk to respiratory healthcare professionals and respiratory medical team about them and to share their distress. Respiratory healthcare professionals achieved initial success in negotiating and accessing symptoms in COPD patients, especially need education, specifically concerning training on how to identify how symptoms are related to one

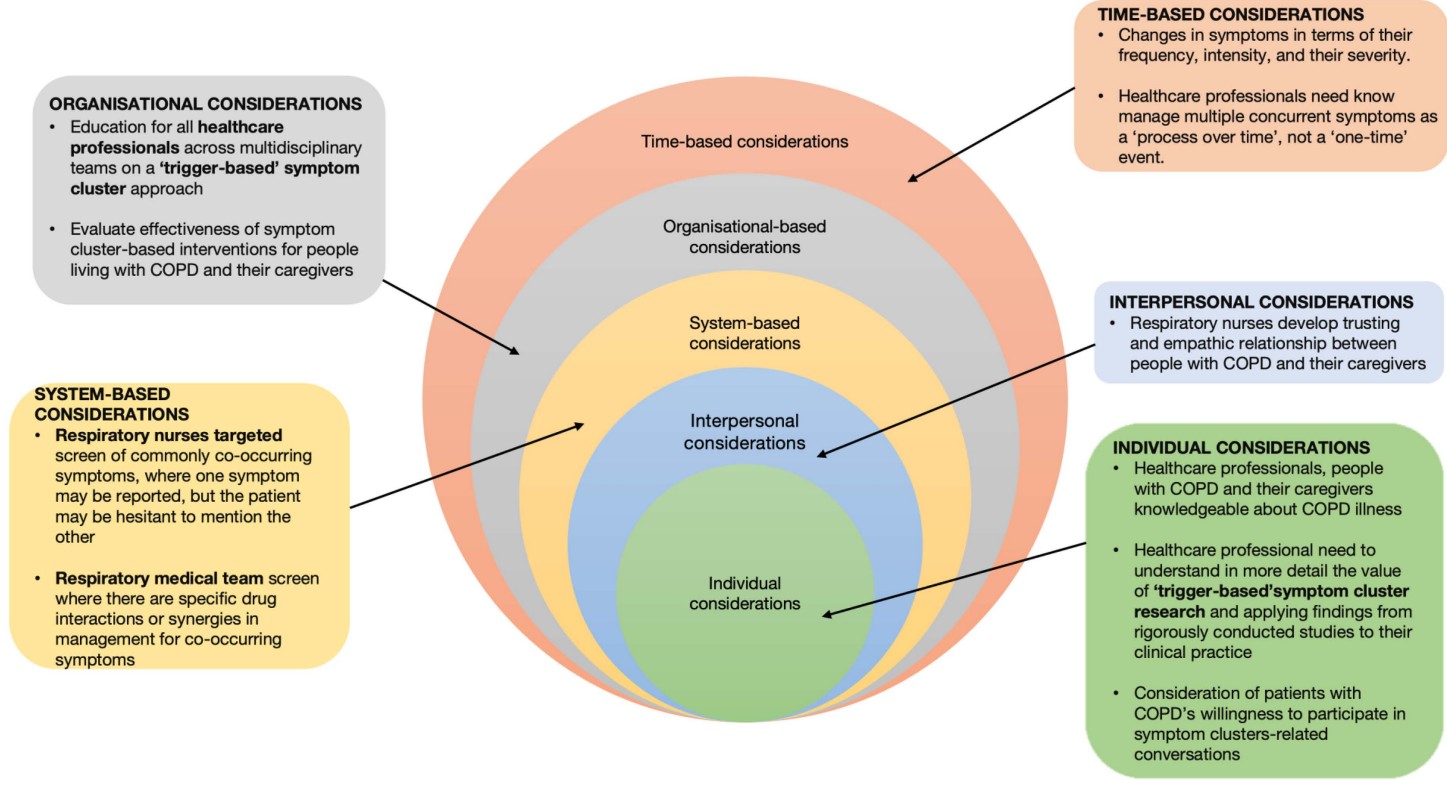

**Fig 3. Multi-level considerations underpinning the management of symptom clusters relevant to people with COPD.**

another and the 'trigger' symptoms that might be present need to know that managing multiple concurrent symptoms is a process over time, not a 'one-time' event. In addition, respiratory healthcare professionals also need to understand in more detail the value of symptom cluster research and need to apply findings from rigorously conducted studies to their clinical routine practice while at the same time developing empathic research relationships between people with COPD and their caregivers.

### 4.3 Relevance to future Research

This current study data represents only a 'snapshot' of participants' lives concerning symptom clusters associated with COPD: their situation and experiences may have provided different results if another time frame had been chosen or if subsequent interviews had been conducted. This design did not permit us to explore the changes in the meanings participants gave to their symptom clusters. Henderson and colleagues have shown that many patients living with life-threatening illnesses are, in principle, agreeable to repeat assessments or interviews in palliative care settings [91]. A method to achieve repeat interviews can be achieved through a good initial relationship between interviewer and participant [87]. Longitudinal research has shown this to be feasible in achieving this [92–94]. Research among those with COPD should follow suit.

## 5 Conclusion

This exploratory qualitative study provides valuable insights into how people living with COPD perceive and interpret their lived experiences with symptom clusters. Participants reported the negative impacts of symptom clusters on their physical,

social and mental well-being. Our findings enrich the existing literature on the strategic management COPD participants adopt to relieve their symptoms. The early identification of COPD participants with symptom clusters and recognition of the central role that 'trigger symptoms' play in experiences of multiple concurrent symptoms may provide insights for the future development of 'trigger-based' symptom cluster interventions. Finally, healthcare providers must become more attuned to value COPD participants' narratives or the meanings of how they comprehend and accommodate their illness. In addition, Hodson et al. [90] have suggested the PREM-C9, a participants-reported experience measure instrument of participants living with COPD, should be used in routine practice to aid healthcare providers in understanding the participants' narratives and to form participants' prioritised goals in co-designed management programmes.

## Supporting information

**S1 File. COREQ checklist.**
(PDF)

**S2 File. Summary of methodological approaches and their roles in this study.**
(DOCX)

## Acknowledgments

The authors wish to thank all patients who enrolled in this study for helping us in the research.

## Author contributions

**Conceptualization:** Fei Fei, Richard J Siegert, Jonathan Koffman.

**Data curation:** Fei Fei, Xiaohan Zhang.

**Formal analysis:** Fei Fei, Xiaohan Zhang.

**Funding acquisition:** Fei Fei.

**Investigation:** Fei Fei.

**Methodology:** Fei Fei, Richard J Siegert, Jonathan Koffman.

**Supervision:** Richard J Siegert, Jonathan Koffman.

**Writing – original draft:** Fei Fei, Xiaohan Zhang.

**Writing – review & editing:** Fei Fei, Richard J Siegert, Jonathan Koffman.

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
