## [Decision Letter · Decision Letter 0]

11 Jan 2026

PONE-D-25-13828Triggered chain reaction: The meanings of symptom clusters for patients with chronic obstructive pulmonary disease: A cross-sectional qualitative studyPLOS One

Dear Dr. Fei,

Thank you for submitting your manuscript to PLOS ONE. After careful consideration, we feel that it has merit but does not fully meet PLOS ONE’s publication criteria as it currently stands. Therefore, we invite you to submit a revised version of the manuscript that addresses the points raised during the review process.

If applicable, we recommend that you deposit your laboratory protocols in protocols.io to enhance the reproducibility of your results. Protocols.io assigns your protocol its own identifier (DOI) so that it can be cited independently in the future. For instructions see: https://journals.plos.org/plosone/s/submission-guidelines#loc-laboratory-protocols. Additionally, PLOS ONE offers an option for publishing peer-reviewed Lab Protocol articles, which describe protocols hosted on protocols.io. Read more information on sharing protocols at . Additionally, PLOS ONE offers an option for publishing peer-reviewed Lab Protocol articles, which describe protocols hosted on protocols.io. Read more information on sharing protocols at https://plos.org/protocols?utm_medium=editorial-email&utm_source=authorletters&utm_campaign=protocols..

We look forward to receiving your revised manuscript.

Kind regards,

Taiwo Opeyemi Aremu, MD, MPH, PhD

Academic Editor

PLOS One

**Journal Requirements:**

- DOI: 10.1371/journal.pone.0265861

- http://dx.doi.org/10.1016/j.ejon.2014.02.004 1462-3889/

In your revision ensure you cite all your sources (including your own works), and quote or rephrase any duplicated text outside the methods section. Further consideration is dependent on these concerns being addressed.

“China Scholarship Council (grant numbers 201806170005).”

4. Please note that funding information should not appear in any section or other areas of your manuscript. We will only publish funding information present in the Funding Statement section of the online submission form. Please remove any funding-related text from the manuscript.

5. In the online submission form you indicate that your data is not available for proprietary reasons and have provided a contact point for accessing this data. Please note that your current contact point is a co-author on this manuscript. According to our Data Policy, the contact point must not be an author on the manuscript and must be an institutional contact, ideally not an individual. Please revise your data statement to a non-author institutional point of contact, such as a data access or ethics committee, and send this to us via return email. Please also include contact information for the third party organization, and please include the full citation of where the data can be found.

6. PLOS requires an ORCID iD for the corresponding author in Editorial Manager on papers submitted after December 6th, 2016. Please ensure that you have an ORCID iD and that it is validated in Editorial Manager. To do this, go to ‘Update my Information’ (in the upper left-hand corner of the main menu), and click on the Fetch/Validate link next to the ORCID field. This will take you to the ORCID site and allow you to create a new iD or authenticate a pre-existing iD in Editorial Manager.

Reviewers' comments:

Reviewer's Responses to Questions

**Comments to the Author**

1. Is the manuscript technically sound, and do the data support the conclusions?

Reviewer #1: Yes

Reviewer #2: Yes

2. Has the statistical analysis been performed appropriately and rigorously? 

Reviewer #1: Yes

Reviewer #2: N/A

3. Have the authors made all data underlying the findings in their manuscript fully available?

Reviewer #1: Yes

Reviewer #2: Yes

4. Is the manuscript presented in an intelligible fashion and written in standard English?

Reviewer #1: Yes

Reviewer #2: Yes

5. Review Comments to the Author

Reviewer #1: Title: Triggered chain reaction: The meanings of symptom clusters for patients with chronic obstructive pulmonary disease: A cross-sectional qualitative study

Thank you for the opportunity to review this manuscript. There are several novel concepts and ideas that are relevant and practical. The follow are my suggestions, comments and suggestions for clarification/clarity and flow for the reader.

Introduction:

Suggest updating references #5-9 (GOLD and WHO reference are a good starting point)

Suggest updating information/references regarding large data sets reference #10

Consider including the origin of ‘system cluster’ from published literature to clarify ‘phenomenon’.

Page 5, line 3. Suggest adding or including Nursing research as the origin of current research. Could also include other areas of research in this area based on current research outside of medical applications (or just state this manuscript/study is only for medical/COPD purposes earlier), such as

Page 7, lines: 5-9. Clarify how the four (4) questions were developed?

Page 8, line 14, suggest incorporating/using consistent term of biographical disruption or biographical theory.

Page 9, line 6, For consistency, flow and clarity who are the ‘Participants’ (patients?) based on department or inclusion criteria.

Page 9, line 8. Clarify or include how TOUS will be used in the study along with the ‘strategy mechanism’ of biographical distribution.

Methods:

Page 10, line 6. Clarify who collected the initial patients or participants criteria: PFT’s, etc.

Page 10, lines 6&7, Clarify and for consistency reconcile ‘participants were recruited from’.....and subheading 2.4 includes Recruitment and sampling for flow.

Page 11, line 1, Clarify the clinician(s), healthcare professionals who identified potential candidates for the study that lead to page 11, line 1, study criteria identification.

Due to the number of theoretical approaches, types of qualitative approaches/processes (descriptive qualitative approach, analysis, etc.), Consolidated Criteria for Reporting Qualitative Research (COREQ), GOLD guidelines/standards, Semi-structured interview topic guide, Framework approach, NVivo, framework matrix, Quality criteria (brief summary of table),

Data collection:

Page 11. Was a method of triangulation or a similar approach utilized to support or clarify a participant's responses.

Page 14, Clarify how ‘strategy mechanism’ of biographical distribution used in addition to or in combination with TOUS- i can see/read this is mentioned in Table 2, suggest mentioning this earlier in this section as only TOUS seems to be mentioned.

Data analysis:

Page 15, line 9, Table 2, consider incorporating how ‘strategy mechanism’ of biographical distribution and TOUS aided in the rigor of the analysis.

Discussion:

For Discussion section and the entire manuscript- in addressing frontline healthcare providers/professionals, all health care professionals contributing and working for the COPD patients/clients in all settings including ‘Respiratory Nursing’. In addition, patient care settings, -hospital, clinic, subacute, long-term rehabilitation, cardiopulmonary rehabilitation, And healthcare professionals, physicians, medical training, respiratory therapist, emergency department personnel, clinics, home care assistants where nursing aids, medical assistants, etc. These healthcare professionals may be the first to see ‘system clusters’ (signs and symptoms) and ‘triggered chain reaction in addition to the setting where they are first seen.

‘Symptoms clusters’ and trigger-based' should be added as a part of patient assessment/physical exam for these patients/clients.

Conclusion:

Consider another possible reason for declining the interviews might be time as the interviews were conducted following treatment, too tired, no time.

Add/incorporate ‘trigger’ concept along with ‘symptom clusters’ in manuscript for clarity and consistency with title - examples: in Figures 2, 3,

Figure 1., clarify 2- ’need to do a temporary physical examination’

Clarify how and if physical assessment/physical exam use of ‘signs’ (signs and symptoms) are part of or not part of this approach when objective data/finding of PFT information is used in selection criteria.

Respectfully submitted.

Reviewer #2: I appreciate you submitting this timely qualitative paper. The topic is clinically meaningful, and the manuscript is generally well organized and easy to follow. I appreciate the effort to report the work using COREQ and to present a clear thematic structure.

To move the paper forward, I recommend minor revision, focused on a small set of required reporting and compliance clarifications (no additional data collection is requested):

Please address the following:

A- Replace any “XX” placeholders with the correct approval number(s) and approving bodies, and ensure the ethics information is consistent across the manuscript and submission fields.

B- The manuscript refers to both written and verbal consent. Please clarify clearly whether consent was written, verbal, or both—and specify what each covered (participation, audio-recording, access to records if applicable). Make sure the description is consistent across all sections.

C- As you know, qualitative transcripts may reasonably be restricted for privacy reasons; however, the Data Availability statement should explain the access route more precisely than “upon request.” Please revise the statement to specify what can be shared (e.g., de-identified excerpts, codebook/framework matrix) and, if transcripts are restricted, provide a clear controlled-access mechanism (e.g., institutional/ethics contact and criteria for access).

D- Please briefly explain how the purposive sampling plan was operationalized (how participants were approached/selected) and clarify what is meant by “clinical interview and cognitive assessment” (what tool or process was used and how it informed eligibility).

E- Please add a short note on transcription accuracy checks and how translation of quotes/extracts was handled to preserve meaning.

Overall, the study is promising, and with these focused clarifications, it will be much stronger and more compliant with journal requirements.

6. PLOS authors have the option to publish the peer review history of their article (what does this mean?). If published, this will include your full peer review and any attached files.). If published, this will include your full peer review and any attached files.

.

Reviewer #1: No

Reviewer #2: No

---

## [Author Response · Author response to Decision Letter 1]

30 Mar 2026

Please refers to the Response Letter.

---

## [Decision Letter · Decision Letter 1]

15 Apr 2026

Triggered chain reaction: The meanings of symptom clusters for patients with chronic obstructive pulmonary disease: A cross-sectional qualitative study

PONE-D-25-13828R1

Dear Dr. Fei,

We’re pleased to inform you that your manuscript has been judged scientifically suitable for publication and will be formally accepted for publication once it meets all outstanding technical requirements.

An invoice will be generated when your article is formally accepted. Please note, if your institution has a publishing partnership with PLOS and your article meets the relevant criteria, all or part of your publication costs will be covered. Please make sure your user information is up-to-date by logging into Editorial Manager at Editorial Manager® and clicking the ‘Update My Information' link at the top of the page. For questions related to billing, please contact  and clicking the ‘Update My Information' link at the top of the page. For questions related to billing, please contact billing support..

Kind regards,

Taiwo Opeyemi Aremu, MD, MPH, PhD

Academic Editor

PLOS One

Additional Editor Comments (optional):

Reviewers' comments:

Reviewer's Responses to Questions

**Comments to the Author**

1. If the authors have adequately addressed your comments raised in a previous round of review and you feel that this manuscript is now acceptable for publication, you may indicate that here to bypass the “Comments to the Author” section, enter your conflict of interest statement in the “Confidential to Editor” section, and submit your "Accept" recommendation.

Reviewer #1: All comments have been addressed

Reviewer #2: All comments have been addressed

2. Is the manuscript technically sound, and do the data support the conclusions?

Reviewer #1: Yes

Reviewer #2: Yes

3. Has the statistical analysis been performed appropriately and rigorously? 

Reviewer #1: Yes

Reviewer #2: N/A

4. Have the authors made all data underlying the findings in their manuscript fully available?

Reviewer #1: Yes

Reviewer #2: Yes

5. Is the manuscript presented in an intelligible fashion and written in standard English?

Reviewer #1: Yes

Reviewer #2: Yes

6. Review Comments to the Author

Reviewer #1: I appreciate the updates and responses to the Clarifications, updates, and suggestions. The flow and readability are improved and make for those who are new (and those in this area of practice) to this area of research and outcomes something that can be utilized to improve their practice and processes.

Reviewer #2: The authors have responded thoroughly and constructively to all comments raised in the first round of review. All editorial requirements have been addressed, including the replacement of ethics approval placeholders, clarification of consent procedures, revision of the data availability statement to specify a controlled-access mechanism, and the addition of transcription and translation transparency.

7. PLOS authors have the option to publish the peer review history of their article (what does this mean?). If published, this will include your full peer review and any attached files.). If published, this will include your full peer review and any attached files.

.

Reviewer #1: No

Reviewer #2: No

---

## [Editor Report · Acceptance letter]

PONE-D-25-13828R1

PLOS One

Dear Dr. Fei,

I'm pleased to inform you that your manuscript has been deemed suitable for publication in PLOS One. Congratulations! Your manuscript is now being handed over to our production team.

Kind regards,

on behalf of

Dr. Taiwo Opeyemi Aremu

Academic Editor

PLOS One